# Fabrication and Microwave Absorption Properties of Core-Shell Structure Nanocomposite Based on Modified Anthracite Coal

**DOI:** 10.3390/nano13121836

**Published:** 2023-06-10

**Authors:** Xiaomei Zhang, Baitong Zhou, Xiang Li, Runhua Chen, Chen Ma, Wenhua Chen, Guohua Chen

**Affiliations:** 1Luoyang Ship Material Research Institute, Luoyang 471003, China; 2Science and Technology on Marine Corrosion and Protection Laboratory, Luoyang 471003, China; 3College of Materials Science and Engineering, Huaqiao University, Xiamen 361021, China; zbt@hqu.edu.cn (B.Z.); chenma@hqu.edu.cn (C.M.); scucwh@163.com (W.C.)

**Keywords:** anthracite coal, Diels-Alder reaction, microwave absorbing performance, core–shell structure

## Abstract

Microwave-absorbing materials have attracted extensive attention due to the development of electronic countermeasures. In this study, novel nanocomposites with core–shell structures based on the core of Fe-Co nanocrystals and the shell of furan methylamine (FMA)-modified anthracite coal (Coal-F) were designed and fabricated. The Diels-Alder (D-A) reaction of Coal-F with FMA creates a large amount of aromatic lamellar structure. After the high-temperature treatment, the modified anthracite with a high degree of graphitization showed an excellent dielectric loss, and the addition of Fe and Co effectively enhanced the magnetic loss of the obtained nanocomposites. In addition, the obtained micro-morphologies proved the core–shell structure, which plays a significant role in strengthening the interface polarization. As a result, the combined effect of the multiple loss mechanism promoted a remarkable improvement in the absorption of incident electromagnetic waves. The carbonization temperatures were specifically studied through a setting control experiment, and 1200 °C was proved to be the optimum parameter to obtain the best dielectric loss and magnetic loss of the sample. The detecting results show that the 10 wt.% CFC-1200/paraffin wax sample with a thickness of 5 mm achieves a minimum reflection loss of −41.6 dB at a frequency of 6.25 GHz, indicating an excellent microwave absorption performance.

## 1. Introduction

Along with the development of electronic technology, the problem of electromagnetic pollution has become more and more serious [1,2]. Therefore, the research of electromagnetic shielding and microwave-absorbing materials has attracted extensive attention [3]. An ideal microwave absorbing material must achieve as much electromagnetic wave as possible into the material to be consumed and a very small amount of electromagnetic wave to be reflected [4]. Dielectric materials, especially carbon materials, show outstanding advantages due to their low density and excellent electrical properties [5]. In this regard, a series of carbon materials with different forms, including conductive carbon black [6], carbon nanotubes [7], graphite [8], and graphene [9], have been investigated.

Coal, as the raw material of different kinds of carbon materials, is the most abundant, widely distributed, and cheapest traditional energy source in the world [10]. It has been used in electricity generation, building materials fields, and steam engine vehicles [11,12]. However, the current application of coal is relatively simple and low-cost. How to improve the value of conventional coal is still a challenging problem. Many works focus on the coal chemical industry [13], graphitization of coal [14,15], and nanostructure transformation of coal [16,17]. Nevertheless, these processes always show a low utilization conversion rate, high energy consumption, and relatively high pollution. Therefore, it is necessary to pay attention to the clean utilization of the coal industry chain and gradually develop a high-efficiency, clean and high value-added coal industry to replace the high-intensity and low-level coal industry.

Coal has a typical porous structure [18]. It has many oxygen-containing functional groups and graphite microcrystalline structures [19], which can provide dipole polarization and dielectric loss to promote microwave absorption. Therefore, coal shows certain potential application as a microwave absorbing material. In addition, coal is a typical example of a highly heterogeneous material with chemical properties, containing complex organic structures and macromolecular networks, which are connected by different covalent and noncovalent bonds [20,21,22]. Compared with general polymer materials, coal has complexity, diversity, and heterogeneity. Research shows that coal can react with some organic substances with conjugated diene structure (furan and its derivatives, cyclopentadiene and its derivatives, etc.) or dienes (maleic acid and its derivatives, vinyl ketones, etc.) by the D-A reaction [23,24,25]. Huang [26] found that when the Diels-Alder (D-A) reaction occurs between furan methylamine (FMA) and coal, the noncovalent bond force in coal is weakened, and the microcrystals in coal are not destroyed. This reaction makes it easier for coal to change to a regular structure via heat treatment, providing more opportunities for functional modification.

Compared with a conventional single structure, a core–shell structure has many advantages in microwave absorption effect [27]. Such a structure could combine the dielectric loss, magnetic loss, and impedance matching, leading to an outstanding microwave absorption performance [28]. Researchers have put forward many kinds of methods to obtain core–shell structures [29,30]. In addition, Li [31] used graphene and Fe-Co nanocrystals as the shell and core to fabricate nanospheres and provided a clear structure model. The obtained nanospheres greatly improved the Fenton-like catalytic performance of the obtained composite. This material system contains carbon and magnetic metal, both two components are potential microwave absorption materials [32]. If the cheap anthracite can be used to replace expensive graphene and build the core–shell structure, the composites could have more chance for industrial application. In addition, the combination of coal and magnetic metal materials to prepare high-performance microwave absorbing composites is of great significance for the high-value application of coal.

To the best of our knowledge, few studies have been carried out that take advantage of coal in microwave absorption fields. In this paper, anthracite macromolecular skeleton (Coal-F) was prepared through D-A reaction. Based on the obtained Coal-F, Fe-Co @ Coal-F nanocomposites (CFCs) with core–shell structures were fabricated. Control experiments were set to specifically study the effect of treating temperatures on the degree of carbonization. Micro-morphologies and element distribution were investigated to prove the core–shell structure of the obtained composites. The porous structure of the sample was analyzed according to the pore size distribution detected through Brunner-Emmet-Teller (BET). XRD, Raman, and temperature-programmed oxidation (TPO) were used to evaluate the degree of carbonization. In addition, the microwave absorption performance of the obtained sample was characterized by measuring electromagnetic parameters, and the microwave absorption mechanism was discussed with the obtained characteristic results.

## 2. Materials and Methods

### 2.1. Materials

FeCl_2_ 4H_2_O, CoCl_2_ 6H_2_O, K_3_(Co(CN)_6_), and dilute hydrochloric acid were brought from Sinopharm Chemical Reagent Co., Ltd. (Shanghai, China), Furan methylamine (FMA), ethanol, and chloroform were provided by Aladdin (Shanghai, China). Anthracite coal was produced at the Tianhushan Coal Mine in Fujian Province (Quanzhou, Fujian Province, China).

### 2.2. Preparation of Coal-F and CFC-T

The pristine coal powders with millimeter scale were smashed, ground, and screened. The obtained powders with micrometer scale were further treated by ball-milling, leach, filtration, and drying. Then, the purified coal with nanometer scale can be prepared and readied for chemical treatment. A total of 3 g of purified coal was mixed in 3 mL of furan methylamine (FMA), and the mixture was heated to 100 °C in a nitrogen atmosphere for 14 days. The obtained reaction product was washed with distilled water, ethanol, and chloroform, respectively, to obtain Coal-F [26]. Then, 1 g Coal-F, 0.99 g FeCl_2_ 4H_2_O, and 1.07 g CoCl_2_ 6H_2_O were subjected to high-frequency ultrasound in 50 mL deionized water for 2 min to obtain the mixed solution. In total, 1.66 g of K_3_(Co(CN)_6_) was added and stirred in 50 mL of deionized water, then slowly poured in the above-mixed solution. After stirring for 30 min, the obtained solution was left to stand for 24 h. The mixture was freeze-dried to obtain a powder, then the excess metal ions were washed away with diluted hydrochloric acid and dried for 6 h. The dried samples were placed in a tubular furnace and heated to 800 °C, 1000 °C, and 1200 °C, respectively, at the heating rate of 5 °C/min in a nitrogen atmosphere and kept for 1 h for carbonization. Finally, after carbonization, the mixture was obtained and named CFC-T, where T represented the carbonization temperature.

### 2.3. Characterization and Testing

The micro-morphologies of nanocomposites were observed through a scanning electron microscope (SEM, FE-SEM, SU8010, Tokyo, Japan). The core–shell structure and element distribution were detected by transmission electron microscopy (TEM, Tecnai 12, Eindhoven, The Netherlands). The nanocomposite samples were placed on a carbon membrane, guaranteeing good electroconductibility for detection. The scanning probe micro-Raman system (Alpha300 RS, Germany) and X-ray diffraction (XRD, X’Pert PRO MPD, PANalytical Inc., Westborough, MA, USA) were used to detect and analyze the graphite structure. Thermogravimetry analysis (TGA, NETZSCH STA 449 F3) was used to study the oxidation behavior of the obtained composites. In addition, the pore size distribution of the sample was detected through Automatic Micropore and Chemisorption Analyzer (Micromeritics/3 Flex, USA). The microwave absorption performance was evaluated by Vector Automatic Network Analyzer (Agilent E5071C, Keysight, Santa Clara, CA, USA).

## 3. Results

### 3.1. Micro-Morphology and Structure of Nanocomposites

Temperature is the main factor affecting the carbonization progress of coal [33]. In order to better analyze the connection between the carbonization temperatures and the micro-structure, SEM was used to specifically detect the details of the obtained CFCs sample. Figure 1a–c shows the morphologies of CFCs prepared at 800 °C, 1000 °C, and 1200 °C, respectively. It can be observed that the CFC-800 samples are composed of block structures, and no carbon sphere exists. When the temperature increased to 1000 °C, a small number of carbon spheres formed on the block surface. It is worth noting that the number of carbon spheres show a remarkable increasement when the temperature reached 1200 °C. According to the high magnification image shown in Figure 1d, the spherical contour also becomes more distinct, and the diameter of a carbon sphere should be 50–200 nm. The different morphology features prove that high carbonization temperature effectively promotes the formation of carbon spheres and plays a significant role in the formation of the spherical structure.

In order to better analyze the specific micro-structure of the different samples treated at different temperatures, we detected the TEM image shown in Figure 2. Figure 2a–c shows the images of CFC-800 at different magnifications. It can be observed that CFC-800 is mainly a sheet-like structure with a small number of regular carbon nanoribbons interspersed in the middle, with a length of ~100 nm and a width of ~5 nm. Figure 2d–f shows the images of CFC-1000. Hollow carbon spheres and carbon-clad metal structures appear in the samples, and the carbon spheres have thinner walls and lower regularity. When the temperature increased to 1200 °C (Figure 2g–i), the length and width of the carbon nanoribbons and graphite ribbons were greatly improved. The hollow carbon spheres and carbon spheres with carbon-coated metal structures are significantly thicker (about 20 nm). Hollow carbon spheres and carbon-coated metal structures have different pore sizes, but most of them are at the nanometer level. The carbon walls are formed by stacking regularly arranged graphite sheets, and some carbon cavities contain metals. According to the TEM image, a good boning at the interface can be observed, which guarantees the structural stability of the nanocomposites.

Figure 3 shows the TEM mapping image of CFC-1200. According to the fact that the distribution areas of Fe and Co elements almost overlap, it can be inferred that the precursor undergoes in situ thermal reduction after high-temperature treatment in the N_2_ atmosphere. The Fe and Co atoms recombine to form a sphere, and the graphite microcrystals near the sphere curl upward at high temperatures and gradually coat the surface of the Fe-Co sphere. Moreover, with the increase in reaction temperature, the content of carbon spheres increases, and the coating is gradually complete. In addition, the size of the carbon sphere wall and graphite strip also becomes larger.

### 3.2. Porous Structure of Nanocomposites

According to micro-morphologies, the prepared sample shows a remarkable porous structure. To further analyze the pore structure of CFC-800, CFC-1000, and CFC-1200, Brunner-Emmet-Teller (BET) measurements were carried out to detect the pore size distribution of the samples, which is shown in Figure 4. The BET results show that CPC-800 and CPC-1200 samples have micropores, mesopores, and macropores, while CFC-1000 mainly has micropores. Coal itself has a porous structure containing micropores, mesopores, and macropores. When the treated temperature is relatively low, the coal structure of CFC-800 almost stays the same as the original coal structure. With the temperature increasing to 1000 °C and the effect of catalysts, the pore structure melts and forms a graphite-like band structure. When the temperature continues to rise, hollow carbon balls begin to form. Therefore, micropores, mesopores, and macropores coexist in CFC-1200. In addition, Table 1 shows the specific surface area and micropore area of the samples. Both the specific surface area and the micropore area gradually decrease with the increase in carbonization temperature.

### 3.3. Crystal Structure of Nanocomposites

Carbonization temperatures have a great effect on the formation of carbon spheres, according to the aforementioned contents. In addition, carbon atoms can rearrange at high temperatures, which can lead to a change in the crystal structure [34]. As shown in Figure 5, the crystal structure and phase composition of CFC-800, CFC-1000, and CFC-1200 were characterized by the XRD. When reacting at 800 °C for 1 h, no obvious graphite characteristic peak can be observed. When the reaction temperature was increased to 1000 °C, a relatively low peak at 2θ = 26.5° appeared, indicating the gradually improved crystal structure of graphite [35]. As the carbonization temperature increased to 1200 °C, the characteristic peaks of graphite became sharper and more intense, meaning the formation of a relatively perfect crystal structure.

The crystal plane spacing between the neighboring carbon–atom layers of the CFC-1000 and CFC-1200 samples was also calculated to further analyze the crystal structure. Based on Bragg’s law [36] and the obtained XRD patterns, the interlayer spacing of CFC-1000 and CFC-1200 was 0.375 nm and 0.339 nm, respectively. According to the study of the coal graphitization law, coal can be mainly divided into four stages during the process of graphitization: the pre-graphitization stage, the initial graphitization stage, the middle graphitization stage, and the high graphitization stage [37]. With the increase in the graphitization stage, the interlayer spacing of the graphite gradually decreases. The interlayer spacing of 0.335 nm is the interlayer space of ideal graphite crystals [38]. The results obtained show that the sample gradually transforms from amorphous to crystalline carbon with the increase in time and temperature. At 1200 °C, the degree of graphitization approaches the middle graphitization stage.

To further prove the graphitization process of the obtained sample, the degree of graphitization was evaluated through Raman spectroscopy. Two typical peaks at 1350 cm^−1^ and 1580 cm^−1^ can be observed, which are assigned to the D-band and G-band, respectively. The D-band is dependent on the disordered carbon structure and defects, while the G-band is dependent on the perfect graphite structure [39]. Therefore, the intensity ratio of the D-band and G-band (I_D_/I_G_) can always be used to evaluate the degree of graphitization of carbon materials [40]. With the carbonization temperature increasing from 800 °C to 1200 °C, the I_D_/I_G_ gradually increase from 0.73 to 0.84, according to the Raman spectra shown in Figure 6, and the G peak gradually moved from 1576 cm^−1^ to the right to 1586 cm^−1^. According to the three-stage model of the change in the G peak position and I_D_/I_G_ value of the amorphous carbon Raman spectrum with the increase in disorder, it can be determined that the sample is in the transformation process of amorphous carbon to nanocrystalline graphite, and it is similar to nanocrystalline graphite.

The degree of graphitization could significantly affect the oxidation behavior, and temperature-programmed oxidation (TPO) was used to specifically characterize the differences in the CFC-800, CFC-1000, and CFC-1200. These three samples were heated at a heating rate of 5 °C/min using a Thermal Gravimetric Analyzer with pure air as the oxidant. At 1000 °C, the TPO of the three samples was obtained by plotting the first derivative of mass loss against the temperature (Figure 7). It can be observed that the initial oxidation temperature shows an increasing trend with the increase in carbonization temperature. CFC-800 shows the lowest oxidation termination temperature and a relatively wider oxidation temperature range. It is worth noting that the oxidation process of CFC-800 has two stages, which are similar to that of CFC-1000 and indicate the existence of carbon structures with a low and a high degree of graphitization. When the carbonization temperature was increased to 1200 °C, a narrower range of oxidation temperature and only one oxidation stage could be observed. The initial oxidation temperature was also higher. This means a relatively uniform composition and a relatively good crystal structure. The TPO results prove that the high-temperature treatment at 1200 °C can effectively improve the crystallization of amorphous carbon existing in anthracite coal.

### 3.4. Microwave Absorption Performance

In order to further study the microwave absorption performance of porous carbon, the electromagnetic parameters of the sample were characterized by the vector network analyzer (VNA). The reflection loss (RL) of porous carbon with different thicknesses in the frequency range of 2–18 GHz was calculated according to the transmission line theory, and the RL curves are shown in Figure 8, respectively. The content of CFC-800, CFC-1000, and CFC-1200 in paraffin wax was fixed to 10 wt.%. It can be observed that the absorption performance shows a gradually enhancing trend with the increase in carbonization temperature. The minimum reflection loss of the CFC-800/paraffin wax was −16.1 dB with a thickness of 5.5 mm at a frequency of 5.24 GHz. The CFC-1000/paraffin wax with a thickness of 5 mm had a minimum reflection loss of −33.9 dB at a frequency of 6.19 GHz. When the carbonization temperature was increased to 1200 °C, the minimum reflection loss of CFC-1200/paraffin wax reached −41.6 dB at a frequency of 6.25 GHz when the thickness was 5 mm. In addition, the CFC-1200/paraffin with a thickness of 2.5 mm showed a wide effective absorbing bandwidth (EAB, RL < −10 dB) from 11.5 GHz to 16.6 GHz. Both the limit value and the effective absorbing bandwidth result indicate that CFC-1200 possess an excellent microwave absorption performance. This is mainly due to the remarkable improvement in dielectric loss and magnetic loss of CFC-1200 samples with the increase in carbonization temperature. In addition, the positive benefit brought by the increase in the number of solid Fe-Co carbon spheres and hollow carbon spheres is greater than the impact of the reduction in specific surface area and the reduction in micropore and mesoporous structures on the microwave absorption performance.

## 4. Conclusions

In summary, based on the conventional materials of anthracite coal, we designed and fabricated core–shell nanocomposites with a good microwave-absorbing performance. The obtained results proved that the carbonization temperature plays an important role in the micro-structure formation of nanocomposites. Furthermore, the combination of porous characteristics, multi-interface nanohybrid structures, and magnetic loss mechanisms of Fe-Co nanocrystals effectively improve the efficiency of absorbing incident electromagnetic waves. The increase in carbonization temperature also leads to the increase in dielectric loss and magnetic loss of CFC-1200 samples. The positive benefit brought by the formation of solid Fe-Co carbon spheres and hollow carbon spheres is greater than the impact of the reduction of specific surface area and the reduction in micropore and mesoporous structures on the microwave absorption performance. Therefore, the minimum reflection loss of the CFC-1200/paraffin wax sample could reach −41.6 dB at a frequency of 6.25 GHz, indicating an outstanding microwave absorption performance. This work could greatly enhance the application value of cheap anthracite coal and provide a new strategy for the design of new wave-absorbing materials.

## Figures and Tables

**Figure 1 nanomaterials-13-01836-f001:**
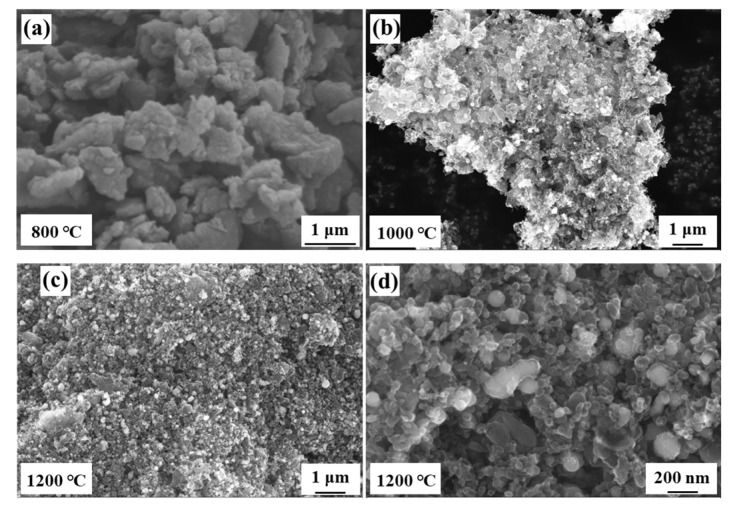
Micro-morphologies of the samples treating at different temperatures. (**a**) CFC-800; (**b**) CFC-1000; (**c**) CFC-1200; (**d**) high magnification image of CFC-1200.

**Figure 2 nanomaterials-13-01836-f002:**
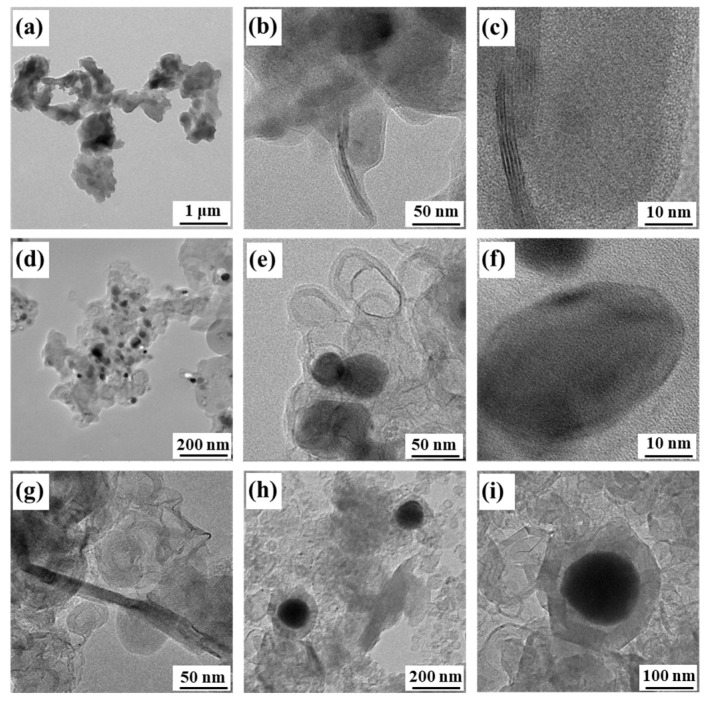
TEM image of CFC-T. (**a**–**c**) CFC-800; (**d**–**f**) CFC-1000; (**g**–**i**) CFC-1200.

**Figure 3 nanomaterials-13-01836-f003:**
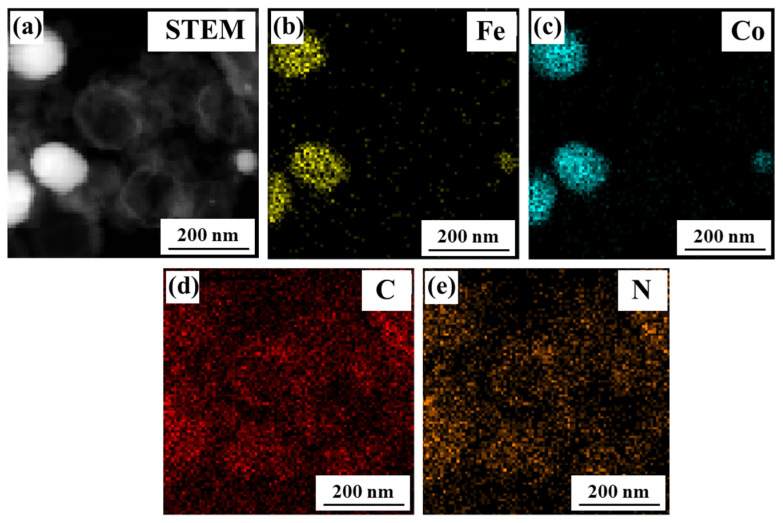
Element distribution of CFC-1200: (**a**) micro-morphologies of sample, (**b**) distribution of Fe, (**c**) distribution of Co, (**d**) distribution of C, (**e**) distribution of N.

**Figure 4 nanomaterials-13-01836-f004:**
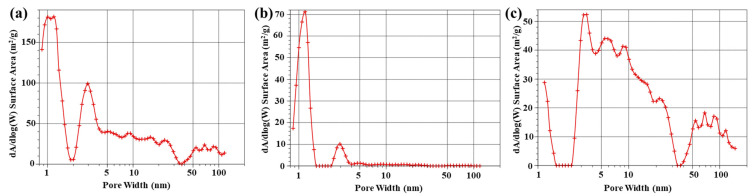
Pore size distribution of sample CFC-T. (**a**) CFC-800; (**b**) CFC-1000; (**c**) CFC-1200.

**Figure 5 nanomaterials-13-01836-f005:**
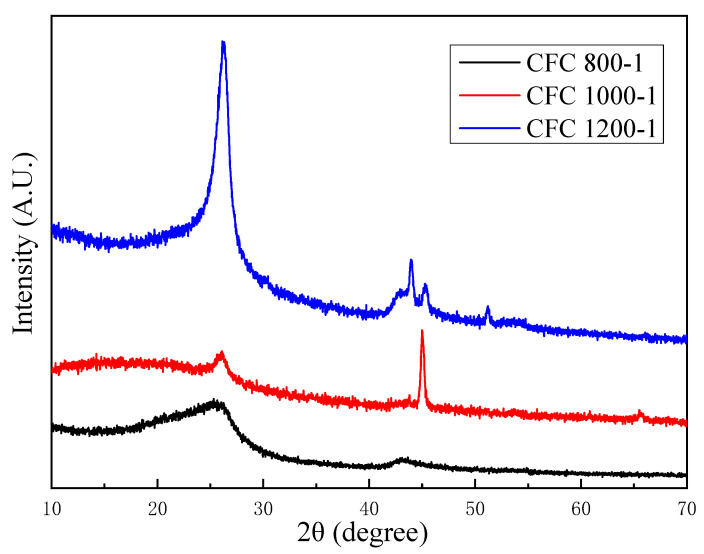
XRD patterns of CFC-800, CFC-1000, CFC-1200 samples.

**Figure 6 nanomaterials-13-01836-f006:**
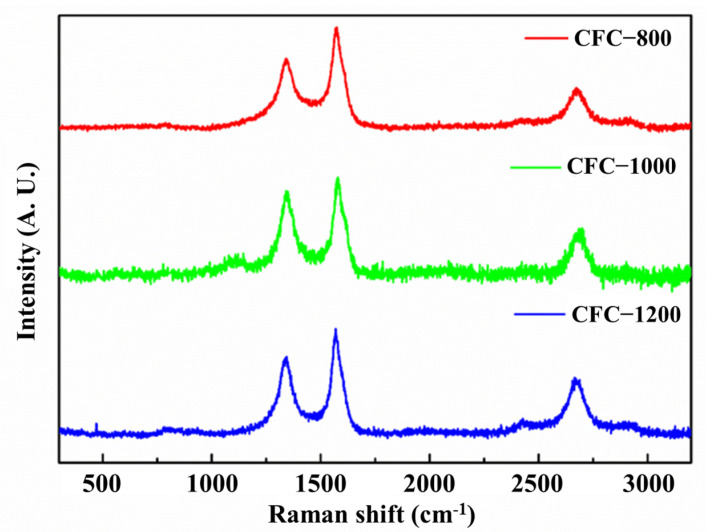
Raman spectra of CFC-800, CFC-1000, CFC-1200 samples.

**Figure 7 nanomaterials-13-01836-f007:**
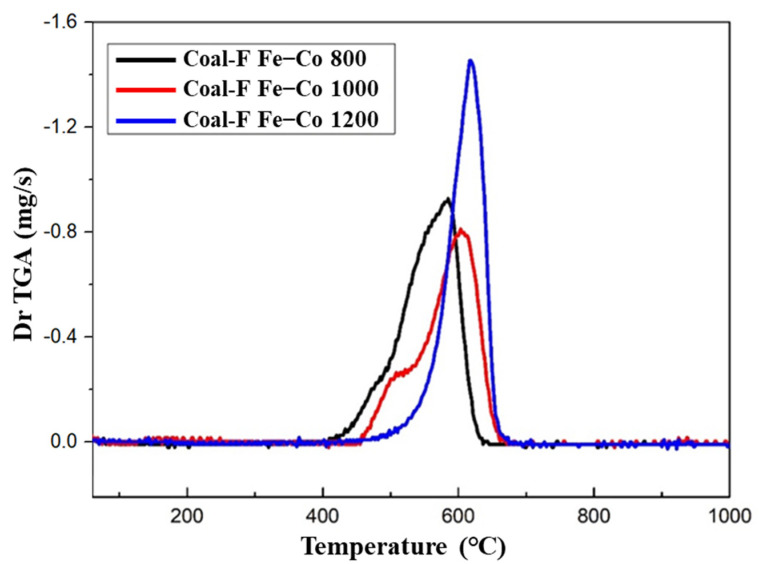
TPO plot of CFC-800, CFC-1000, CFC-1200 samples.

**Figure 8 nanomaterials-13-01836-f008:**
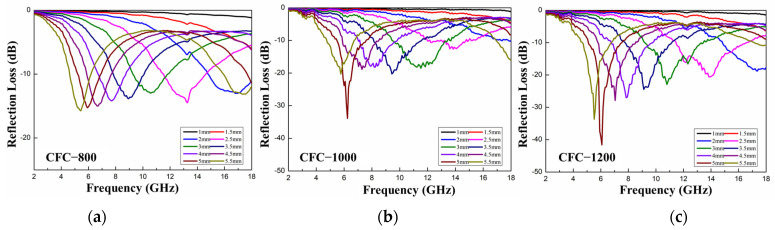
Reflection loss of CFC-T sample treated at different temperatures. (**a**) CFC-800; (**b**) CFC-1000; (**c**) CFC-1200.

**Table 1 nanomaterials-13-01836-t001:** Specific surface area of CPC-X.

CFC-T	BET Surface Area (m^2^/g)	t-Plot Micropore Area (m^2^/g)
CFC-800	199	43.7
CFC-1000	175	33.1
CFC-1200	103	10.43

## Data Availability

Not applicable.

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
