# Peer review of "Fabrication and Microwave Absorption Properties of Core-Shell Structure Nanocomposite Based on Modified Anthracite Coal"

_nanomaterials, 2023, doi:10.3390/nano13121836_

Round 1

Reviewer 1 Report

11)      English check is mandatory.

22)      There is no information describing the preparation of TEM samples, this is crucial to understand: for example, the powders are placed on a carbon net (sample holder)? In this case its mapping would be useless!

33)      What is the size (millimeter, micrometer, nanometer scale) of the pristine coal? How are the powders for chemical treatments with Fe and Co obtained? Generally, the pieces of coal have a mass greater than 3 g. Are the powders made from larger pieces, are they ground or something else? Please, details are needed.

Line 11: “of Fe-CO nanocrystals” is not defined or “CO” is “Co”.

Line 66: “Fe-Co nano crystal”. "Fe-Co Nano Crystal". Please decide whether to use nanocrystals or nanocrystals and keep it in the text.

Line 94: what does “the obtained product” means? The result of a chemical reaction between FMA and Coal or is it only coal heat treated? Please add a reference or more details! 

Line 103: Authors report “Finally, the carbonated mixture was obtained and named CFC-T, where T represented the carbonization temperature.” Generally, chemistry speaking carbonated is referred to the dissolution of CO2 in water or in beverage to produce the carbonation, i.e. the saturation of a liquid with CO2 gas. In other words, it is a term used to describe the dissolution of CO2 gas in water utilizing pressure and temperature. So, carbonization is more appropriate.

Figure 1. It is not strange that the grain size is similar for particles treated at 1000 and 1200 °C? What about the particles treated at 800 °C, if present the effect should be bigger? Why images of sample CFC800 aren’t shown? Are the images shown representative of the whole sample?

Figure 4 a, b and c result low in quality and their axis labels (as well as the scales) are un-readable.

Line 182: “2θ degree” is better than “26.5 ° ”.

Line 189: into “was 0.375 and 0.339 respectively” both distances do not show any unit.

Line 193: into “of 0.3375 can” the unit is missed.

Both graphs of fig. 5 and 6 need A.U. (Arbitrary Unit) on the y axis. Furthermore, also the y axis of figure 7 need for a unit and relative physical characteristic.

Line 194-196: what does graphitization mean? How much carbon has been added to something or more amount of crystalline phase has been reached? Please, could you explain better? If you mean carbonization, is better to maintain this term than graphitization.

The transformation of non-graphitic carbon is generally performed at about 2200-2700 °C [1], is not too low the temperature range used here (800 – 1200 °C) to change the carbon structure? Also considering the period of heating of 1 hour.

I tried to find in literature something about “the graphitization law” (line 190) but nothing was found, please could you provide more details and references about that.

What does “a-C carbon” means? If it stays for amorphous carbon, amorphous-C is better!

Perhaps the criticism that can be made of this work is that no comparison is made.

For example, what are improvements in the response of prepared samples compared to untreated coal (no addition of Fe and Co) or respect to some other material known and routinely used in the field of applications of interest?

[1] Production and Reference Material Harry Marsh, Francisco Rodríguez-Reinoso, in Activated Carbon, 2006

https://www.sciencedirect.com/topics/earth-and-planetary-sciences/graphitization

Author Response

A point-by-point reply to reviewers' comments:

Reviewer 1:

(1) English check is mandatory.

 Reply: We apologize for the poor language of our manuscript. We worked on the manuscript for a long time and the repeated addition and removal of sentences and sections obviously led to poor readability. We have now worked on both language and readability for language corrections. We really hope that the flow and language level have been substantially improved.

(2) There is no information describing the preparation of TEM samples, this is crucial to understand: for example, the powders are placed on a carbon net (sample holder)? In this case its mapping would be useless!

 Reply: Thank you for reviewer’s good suggestion. The TEM sample was place on a carbon membrane that was prepared by spraying carbon materials on an organic membrane. Although the C element on carbon membrane can also be detected in mapping result, the obtained image shown in Figure 3 indicates an obvious gathering characteristic. The element enriched area is the nanocomposites. Therefore, the mapping results can still give us the element distribution information, although the sample was place on carbon membrane.

In order to better describe the preparation process of TEM sample, we added the preparation details in the manuscript. The improved contents are shown below. (Line 112-115 in the manuscript)

The core-shell structure and element distribution were detected by transmission electron microscopy (TEM, Tecnai 12, Holland). The nanocomposite samples were placed on a carbon membrane, guaranteeing the good electroconductibility for detecting.

Figure 3. Element distribution of CFC-1200: (a) micro-morphologies of sample, (b) distribution of Fe, (c) distribution of Co, (d) distribution of C, (e) distribution of N.

(3) What is the size (millimeter, micrometer, nanometer scale) of the pristine coal? How are the powders for chemical treatments with Fe and Co obtained? Generally, the pieces of coal have a mass greater than 3 g. Are the powders made from larger pieces, are they ground or something else? Please, details are needed.

 Reply: Thank you for reviewer’s good suggestion. It is true that the powders were made from large pieces. The pristine coal would be smashed, ground, and screened. The obtained powders with micrometer scale would be further treated by ball-milling, leach, filtration, and dry. Then, the purified coal with nanometer scale can be prepared and ready for chemical treatments with Fe and Co.

In order to better describe the treating process of pristine coal, we added the details in the manuscript. The improved contents are shown below. (Line 94-97 in the manuscript)

The pristine coal powders with millimeter scale were smashed, ground, and screened. The obtained powders with micrometer scale were further treated by ball-milling, leach, filtration, and dry. Then, the purified coal with nanometer scale can be prepared and ready for chemical treatment.

(4) Line 11: “of Fe-CO nanocrystals” is not defined or “CO” is “Co”.

Reply: Thanks to reviewer for reminder. It is a mistake, and it should be “Fe-Co nanocrystals”. We have corrected it in the manuscript. (Line 11 in the manuscript)

(5) Line 66: “Fe-Co nano crystal”. "Fe-Co Nano Crystal". Please decide whether to use nanocrystals or nanocrystals and keep it in the text.

Reply: Thanks to reviewer for reminder. It should be “nanocrystals”, and we have corrected this word in the manuscript. (Line 11 in the manuscript)

(6) Line 94: What does “the obtained product” means? The result of a chemical reaction between FMA and Coal or is it only coal heat treated? Please add a reference or more details!

Reply: Thank you for reviewer’s good suggestion. The obtained product mentioned in the manuscript is the reaction product between FMA and Coal. This reaction has been proved in the reference [26]. In order to make it easier for readers to understand, we have added some details and the reference in the manuscript. The improved contents are shown below.

3 g purified coal was mixed in 3 mL furan methylamine (FMA), and the mixture was heated to 100 ℃ in a nitrogen atmosphere for 14 days. The obtained reaction product was washed with distilled water, ethanol, and chloroform respectively to obtain Coal-F [26]. (Line 97-100 in the manuscript)

[26]Huang, Y.; Chen, S.; Ma, R.; Cheng, Y.; Jin, L.; Chen, G.J.A.C.; Materials, H. Coal-based carbon composite with excellent electromagnetic-shielding properties prepared from modification of coal with D-A reaction. 1-13.

(7) Line 103: Authors report “Finally, the carbonated mixture was obtained and named CFC-T, where T represented the carbonization temperature.” Generally, chemistry speaking carbonated is referred to the dissolution of CO2 in water or in beverage to produce the carbonation, i.e. the saturation of a liquid with CO2 gas. In other words, it is a term used to describe the dissolution of CO2 gas in water utilizing pressure and temperature. So, carbonization is more appropriate.

Reply: Thank you for reviewer’s good suggestion. We have corrected this word in the manuscript. The corrected contents are shown below.

Finally, the mixture after carbonization was obtained and named CFC-T. (Line 108 in the manuscript)

(8) Figure 1. It is not strange that the grain size is similar for particles treated at 1000 and 1200 °C? What about the particles treated at 800 °C, if present the effect should be bigger? Why images of sample CFC800 aren’t shown? Are the images shown representative of the whole sample?

Reply: Thank you for reviewer’s good suggestion. The images of sample CF-800 have been added in the manuscript, and the morphology differences of samples treated at different temperatures are also analyzed. The improved contents are shown below. (Line 128-137 in the manuscript)

Figure 1 (a, b, c) show the morphologies of CFCs prepared at 800 ℃, 1000 ℃ and 1200 ℃ respectively. It can be observed that the CFC-800 samples are composed of block structures, and no carbon sphere exists. When the temperature was increased to 1000 ℃, a small number of carbon sphere formed on the block surface. It is worth noting that the number of carbon sphere show a remarkable increasement when the temperature reaching to 1200 ℃. According to the high magnification image shown in Figure 1 (d), the spherical contour also gets more distinct, and the diameter of carbon sphere should be 50-200 nm. The different morphology features prove that high car-bonization temperature effectively promotes the formation of carbon sphere and plays a significant role in forming the spherical structure.

Figure 1. Micro-morphologies of the samples treating at different temperatures. (a) CFC-800; (b) CFC-1000; (c) CFC-1200; (d) high magnification image of CFC-1200

(9) Figure 4 a, b and c result low in quality and their axis labels (as well as the scales) are un-readable.

Reply: Thank you for reviewer’s good suggestions. Figure 4 has been corrected. The improved figure is shown below. (Line 184 in the manuscript)

Figure 4. Pore size distribution of sample CFC-T. (a) CFC-800; (b) CFC-1000; (c) CFC-1200

(10) Line 182: “2θ degree” is better than “26.5°”.

Reply: Thank you for reviewer’s good suggestion. We have corrected the written form. The corrected contents are shown below. (Line 193 in the manuscript)

While the reaction temperature was increased to 1000 ℃, a relatively low peak at 2θ= 26.5° appears.

(11) Line 189: into “was 0.375 and 0.339 respectively” both distances do not show any unit.

Reply: Thank you for reviewer’s good suggestion. We have added the unit. The improved contents are shown below. (Line 200 in the manuscript)

Based on the Bragg's law and the obtained XRD patterns, the interlayer spacing of CFC-1000 and CFC-1200 was 0.375 nm and 0.339 nm respectively.

(12) Line 193: into “of 0.3375 can” the unit is missed.

Reply: Thank you for reviewer’s good suggestion. We have added the unit. The improved contents are shown below. (Line 204 in the manuscript)

The interlayer spacing of 0.335 nm is the interlayer space of ideal graphite crystal.

(13) Both graphs of fig. 5 and 6 need A.U. (Arbitrary Unit) on the y axis. Furthermore, also the y axis of figure 7 need for a unit and relative physical characteristic.

Reply: Thank you for reviewer’s good suggestion. The A.U. has been added in Figure 5 and Figure 6. The unite is also added in the y axis of Figure 7. The corrected Figures are shown below. (Line 210, 225, 243 in the manuscript)

Figure 5. XRD patterns of CFC-800, CFC-1000, CFC-1200 sample

Figure 6. Raman spectra of CFC-800, CFC-1000, CFC-1200 samples

Figure 7. TPO plot of sample CFC-T.

(14) Line 194-196: what does graphitization mean? How much carbon has been added to something or more amount of crystalline phase has been reached? Please, could you explain better? If you mean carbonization, is better to maintain this term than graphitization.

Reply: Thank you for reviewer’s good questions. Graphitization means the atomic structure of carbon materials changing from the amorphous structure to crystal structure. Therefore, graphitization is different from carbonization. In this study, the transformation from coal to graphite corresponding to the atomic structure changing process. Hence, graphitization could be more appropriate to be used in this manuscript.

(15) The transformation of non-graphitic carbon is generally performed at about 2200-2700 °C [1], is not too low the temperature range used here (800 – 1200 °C) to change the carbon structure? Also considering the period of heating of 1 hour.

Reply: Thank you for reviewer’s good questions. It is true that the transformation temperature of normal coal to graphite would be 2200-2700 ℃. However, the coal used in this study has been modified with FMA. D-A reaction between coal and FMA would reduce the covalent bond force between different carbon atoms, leading to the easily transformation from amorphous form to perfect crystal structure. In addition, the introduction of Fe and Co atoms shows the heterogeneous catalysis effect, which could reduce the transformation activation energy from coal to graphite. Therefore, the transformation of non-graphitic carbon could occur at a relatively low temperature in this study.

(16) I tried to find in literature something about “the graphitization law” (line 190) but nothing was found, please could you provide more details and references about that.

Reply: Thank you for reviewer’s good suggestions. The related references have been added in the manuscript. The corrected contents are shown below. (Line 200-205 in the manuscript)

According to the study of coal graphitization law, coal can be mainly divided into four stages in the process of graphitization: pre-graphitization stage, initial graphitization stage, middle graphitization stage, and high graphitization stage[37]. With the increase of the graphitization stage, the interlayer spacing of graphite gradually decreases. The interlayer spacing of 0.335 nm is the interlayer space of ideal graphite crystal[38].

[37] Zheng Z. HRTEM study on microstructures of coal-based graphite. Acta Mineralogica Sinica 1991, 11, 14-218.

[38] Seehra M S, Pavlovic A S. X-Ray diffraction, thermal expansion, electrical conductivity, and optical microscopy studies of coal-based graphites. Carbon 1993, 31, 557-564.

(17) What does “a-C carbon” means? If it stays for amorphous carbon, amorphous-C is better!

Reply: Thank you for reviewer’s good questions. This word has been changed to amorphous carbon. The corrected contents are shown below. (Line 222 in the manuscript)

It can be determined that the sample is in the transformation process of amorphous carbon to nanocrystalline graphite.

(18) Perhaps the criticism that can be made of this work is that no comparison is made. For example, what are improvements in the response of prepared samples compared to untreated coal (no addition of Fe and Co) or respect to some other material known and routinely used in the field of applications of interest?

Reply: Thank you for reviewer’s good questions. The performance of microwave absorption performance could be evaluated and compared through the reflection loss results of different samples. It can be considered as a good absorption performance in practical application if the reflection loss could be lower than -10 dB. Therefore, the performance of products obtained in this study can be easily evaluated through the above method, and we believe the performance of CFC-1200 sample have certain advantages compared with the conventional microwave-absorption materials. For the comparison experiments, we would do more additional experiments to further improve the performance and achieve a further understanding concerning the microwave absorption mechanism in the future.

Reviewer 2 Report

In this study, novel nanocomposites (CFCs) with core-shell structure based on the core of Fe-CO nanocrystals and the shell of furan methylamine (FMA) modified an- thracite coal (Coal-F) were designed and fabricated. In addition, the obtained micro-morphologies proved the core-shell structure, which plays a significant role in strengthening the interface polarization. 1.  Research related to the time stability of the developed structure is necessary. And conducting research on the impact of high-energy impulses 2.  The development of a mathematical model of the structure relating to the parameters is missing 3.  Descriptions in Figure 8 are fuzzy

Author Response

A point-by-point reply to reviewers' comments:

Reviewer 2:

In this study, novel nanocomposites (CFCs) with core-shell structure based on the core of Fe-CO nanocrystals and the shell of furan methylamine (FMA) modified an- thracite coal (Coal-F) were designed and fabricated. In addition, the obtained micro-morphologies proved the core-shell structure, which plays a significant role in strengthening the interface polarization. 

  1. Research related to the time stability of the developed structure is necessary. And conducting research on the impact of high-energy impulses. 

Reply: Thank you for reviewer’s good suggestions. The core-shell structure was obtained at high temperature, and the graphitization had occurred during thermal treatment. Therefore, the structure of carbon itself is very stable. In addition, the thermal treatment also leads to a pretty strong bond between coal and Fe-Co nanocrystal. The TEM image shown in Figure 2 (i) indicates a good bonding at the interface, no crack can be observed. Therefore, the excellent combination between coal and Fe-Co nanocrystal guarantees the structure and time stability.

In order to describe the stabile performance, we added the description in the manuscript. The improved contents are shown below. (Line 153-155 in the manuscript)

According to the TEM image, a good boning at the interface can be observed, which guarantees the structure stability of the nanocomposites.

For the research concerning the impact of high-energy impulse, this study is not involved with high-energy impulse. Maybe we can try to do some related research in the future.

Figure 2 (i) TEM image of CFC-1200

  1. The development of a mathematical model of the structure relating to the parameters is missing.

Reply: Thank you for reviewer’s good suggestion. The model of the Fe-Co structure has been studied and provided in the previous research. In order to give reader a better understanding to this model, we cite this reference in the manuscript. The modified contents are shown below. (Line 66-69 in the manuscript)

Especially, Li[31] used graphene and Fe-Co nanocrystals as the shell and core to fab-ricate nanospheres, and provided a clear structure model. The obtained nanospheres greatly improved the Fenton-like catalytic performance of the obtained composite.

[31] Li, X.N.; Huang, X.; Xi, S.B.; Miao, S.; Ding, J.; Cai, W.Z.; Liu, S.; Yang, X.L.; Yang, H.B.; Gao, J.J.; et al. Single Cobalt Atoms Anchored on Porous N-Doped Graphene with Dual Reaction Sites for Efficient Fenton-like Catalysis. Journal of the American Chemical Society 2018, 140, 12469-12475.

  1. Descriptions in Figure 8 are fuzzy

Reply: Thank you for reviewer’s good suggestion. The description in Figure 8 has been improved. The corrected contents are shown below. (Line 251-261 in the manuscript)

In order to further study the microwave absorption performance of porous carbon, the electromagnetic parameters of the sample were characterized by the vector net-work analyzer (VNA). According to the transmission line theory, the reflection loss (RL) of porous carbon with different thicknesses in the frequency range of 2-18 GHz was calculated, and the RL curves are shown in Figure 8 respectively. The content of CFC-800, CFC-1000, and CFC-1200 in paraffin wax was fixed to 10 wt.%. It can be ob-served that the absorption performance shows a gradually enhancing trend with the increase of carbonization temperature. The minimum reflection loss of CFC-800/paraffin wax was -16.1 dB with a thickness of 5.5 mm at a frequency of 5.24 GHz. The CFC-1000/paraffin wax with a thickness of 5 mm had a minimum reflection loss of -33.9 dB at a frequency of 6.19 GHz. When the carbonization temperature was increased to 1200 ℃, the minimum reflection loss of CFC-1200/paraffin wax reached to -41.6 dB at a frequency of 6.25 GHz when the thickness is 5 mm. In addition, the CFC-1200/paraffin with the thickness of 2.5 mm showed a wide effective absorbing bandwidth (EAB, RL<-10 dB) from 11.5 GHz to 16.6 GHz. Both the limit value and the effective absorbing bandwidth result indicate that CFC-1200 owns excellent micro-wave absorption performance. This is mainly due to the remarkable improvement of dielectric loss and magnetic loss of CFC-1200 samples with the increase of carboniza-tion temperature. In addition, the positive benefit brought by the increase of the num-ber of solid Fe-Co carbon spheres and hollow carbon spheres is greater than the impact of the reduction of specific surface area and the reduction of micropore and mesopo-rous structure on the microwave absorption performance.

Round 2

Reviewer 1 Report

The authors revised their draft in accordance with the review request and adequately answered each question, consequently the article can be published in its current form.